# X-CAL: Explicit Calibration for Survival Analysis

**Mark Goldstein**[*]
New York University
goldstein@nyu.edu

**Xintian Han**[*]
New York University
xintian.han@nyu.edu

**Aahlad Puli**[*]
New York University
aahlad@nyu.edu

**Adler J. Perotte**
Columbia University
adler.perotte@columbia.edu

**Rajesh Ranganath**
New York University
rajeshr@cims.nyu.edu

## Abstract

Survival analysis models the distribution of time until an event of interest, such as discharge from the hospital or admission to the ICU. When a model's predicted number of events within any time interval is similar to the observed number, it is called *well-calibrated*. A survival model's calibration can be measured using, for instance, distributional calibration (D-CALIBRATION) [Haider et al., 2020] which computes the squared difference between the observed and predicted number of events within different time intervals. Classically, calibration is addressed in post-training analysis. We develop explicit calibration (X-CAL), which turns D-CALIBRATION into a differentiable objective that can be used in survival modeling alongside maximum likelihood estimation and other objectives. X-CAL allows practitioners to directly optimize calibration and strike a desired balance between predictive power and calibration. In our experiments, we fit a variety of shallow and deep models on simulated data, a survival dataset based on MNIST, on length-of-stay prediction using MIMIC-III data, and on brain cancer data from The Cancer Genome Atlas. We show that the models we study can be miscalibrated. We give experimental evidence on these datasets that X-CAL improves D-CALIBRATION without a large decrease in concordance or likelihood.

## 1 Introduction

A core challenge in healthcare is to assess the risk of events such as onset of disease or death. Given a patient's vitals and lab values, physicians should know whether the patient is at risk for transfer to a higher level of care. Accurate estimates of the time-until-event help physicians assess risk and accordingly prescribe treatment strategies: doctors match aggressiveness of treatment against severity of illness. These predictions are important to the health of the individual patient and to the allocation of resources in the healthcare system, affecting all patients.

Survival Analysis formalizes this risk assessment by estimating the conditional distribution of the *time-until-event* for an outcome of interest, called the failure time. Unlike supervised learning, survival analysis must handle datapoints that are *censored*: their failure time is not observed, but bounds on the failure time are. For example, in a 10 year cardiac health study [Wilson et al., 1998, Vasan et al., 2008], some individuals will remain healthy over the study duration. Censored points are informative, as we can learn that someone's physiology indicates they are healthy-enough to avoid onset of cardiac issues within the next 10 years.

A *well-calibrated* survival model is one where the predicted number of events within any time interval is similar to the observed number [Pepe and Janes, 2013]. When this is the case, event

---

[*]Equal Contribution

probabilities can be interpreted as risk and can be used for downstream tasks, treatment strategy, and human-computable risk score development [Sullivan et al., 2004, Demler et al., 2015, Haider et al., 2020]. Calibrated conditional models enable accurate, individualized prognosis and may help prevent giving patients misinformed limits on their survival, such as 6 months when they would survive years. Poorly calibrated predictions of time-to-event can misinform decisions about a patient's future.

Calibration is a concern in today's deep models. Classical neural networks that were not wide or deep by modern standards were found to be as calibrated as other models after the latter were calibrated (boosted trees, random forests, and SVMs calibrated using Platt scaling and isotonic regression) [Niculescu-Mizil and Caruana, 2005]. However, deeper and wider models using batchnorm and dropout have been found to be overconfident or otherwise miscalibrated [Guo et al., 2017]. Common shallow survival models such as the Weibull Accelerated Failure Times (AFT) model may also be miscalibrated [Haider et al., 2020]. We explore shallow and deep models in this work.

Calibration checks are usually performed post-training. This approach decouples the search for a good predictive model and a well-calibrated one [Song et al., 2019, Platt, 1999, Zadrozny and Elkan, 2002]. Recent approaches tackle calibration in-training via alternate loss functions. However, these may not, even implicitly, optimize a well-defined calibration measure, nor do they allow for explicit balance between prediction and calibration [Avati et al., 2019]. Calibration during training has been explored recently for binary classification [Kumar et al., 2018]. Limited evaluations of calibration in survival models can be done by considering only particular time points: *this model is well-calibrated for half-year predictions*. Recent work considers D-CALIBRATION [Haider et al., 2020], a holistic measure of calibration of time-until-event that measures calibration of *distributions*.

In this work, we propose to improve calibration by augmenting traditional objectives for survival modeling with a differentiable approximation of D-CALIBRATION, which we call explicit calibration (X-CAL). X-CAL is a plug-in objective that reduces obtaining good calibration to an optimization problem amenable to data sub-sampling. X-CAL helps build well-calibrated versions of many existing models and controls calibration *during* training. In our experiments [2], we fit a variety of shallow and deep models on simulated data, a survival dataset based on MNIST, on length-of-stay prediction using MIMIC-III data, and on brain cancer data from The Cancer Genome Atlas. We show that the models we study can be miscalibrated. We give experimental evidence on these datasets that X-CAL improves D-CALIBRATION without a large decrease in concordance or likelihood.

## 2   Defining and Evaluating Calibration in Survival Analysis

Survival analysis models the time $\mathbf{t} > 0$ until an event, called the failure time. $\mathbf{t}$ is often assumed to be conditionally distributed given covariates $\mathbf{x}$. Unlike typical regression problems, there may also be censoring times $\mathbf{c}$ that determine whether $\mathbf{t}$ is observed. We focus on right-censoring in this work, with observations $(u, \delta, x)$ where $\mathbf{u} = \min(\mathbf{t}, \mathbf{c})$ and $\boldsymbol{\delta} = \mathbb{1}\left[\mathbf{t} < \mathbf{c}\right]$. If $\delta = 1$ then $u$ is a failure time. Otherwise $u$ is a censoring time and the datapoint is called *censored*. Censoring times may be constant or random. We assume censoring-at-random: $\mathbf{t} \perp\!\!\!\perp \mathbf{c} \mid \mathbf{x}$.

We denote the joint distribution of $(\mathbf{t}, \mathbf{x})$ by $P$ and the conditional cumulative distribution function (CDF) of $\mathbf{t} \mid \mathbf{x}$ by $F$ (sometimes denoting the marginal CDF by $F$ when clear). Whenever distributions or CDFs have no subscript parameters, they are taken to be true data-generating distributions and when they have parameters $\theta$ they denote a model. We give more review of key concepts, definitions, and common survival analysis models in Appendix A.

### 2.1   Defining Calibration

We first establish a common definition of calibration for binary outcome. Let $\mathbf{x}$ be covariates and let $\mathbf{d}$ be a binary outcome distributed conditional on $\mathbf{x}$. Let them have joint distribution $P(\mathbf{d}, \mathbf{x})$. Define $\texttt{risk}_\theta(x)$ as the modeled probability $P_\theta(\mathbf{d} = 1 \mid x)$, a deterministic function of $x$. Pepe and Janes [2013] define calibration as the condition that

$$\mathbb{P}(\mathbf{d} = 1 \mid \texttt{risk}_\theta(\mathbf{x}) = r) =\approx r. \tag{1}$$

That is, the frequency of events is $r$ among subjects whose modeled risks are equal to $r$. For a survival problem with joint distribution $P(\mathbf{t}, \mathbf{x})$, we can define risk to depend on an observed failure time

instead of the binary outcome $\mathbf{d} = 1$. With $F_\theta$ as the model CDF, the definition of risk for survival analysis becomes $\mathtt{risk}_\theta(t, x) = F_\theta(t \mid x)$, a deterministic function of $(t, x)$. Then perfect calibration is the condition that, for all sub-intervals $I = [a, b]$ of $[0, 1]$,

$$\mathbb{P}(\mathtt{risk}_\theta(\mathbf{t}, \mathbf{x}) \in I) = \underset{P(\mathbf{t}, \mathbf{x})}{\mathbb{E}} \, \mathbb{1}\left[F_\theta(\mathbf{t} \mid \mathbf{x}) \in I\right] = |I|. \tag{2}$$

This is because, for continuous $F$ (an assumption we keep for the remainder of the text), CDFs transform samples of their own distribution to $\mathrm{Unif}(0, 1)$ variates. Thus, when model predictions are perfect and $F_\theta = F$, the probability that $F_\theta(\mathbf{t} \mid \mathbf{x})$ takes a value in interval $I$ is equal to $|I|$. Since the expectation is taken over $\mathbf{x}$, the same holds when $F_\theta(t \mid x) = F(t)$, the true marginal CDF.

## 2.2 Evaluating Calibration

Classical tests and their recent modifications assess calibration of survival models for a particular time of interest $t^*$ by comparing observed versus modeled event frequencies [Lemeshow and Hosmer Jr, 1982, Grønnesby and Borgan, 1996, D'agostino and Nam, 2003, Royston and Altman, 2013, Demler et al., 2015, Yadlowsky et al., 2019]. They apply the condition in Equation (1) for the classification task $\mathbf{t} < t^* \mid \mathbf{x}$. These tests are limited in two ways 1) it is not clear how to combine calibration assessments over the entire range of possible time predictions [Haider et al., 2020] and 2) they answer calibration in a rigid yes/no fashion with hypothesis testing. We briefly review these tests in Appendix A.

**D-CALIBRATION**  Haider et al. [2020] develop distributional calibration (D-CALIBRATION) to test the calibration of conditional survival *distributions* across all times. D-CALIBRATION uses the condition in Equation (2) and checks the extent to which it holds by evaluating the model conditional CDF on times in the data and checking that these CDF evaluations are uniform over $[0, 1]$. This uniformity ensures that observed and predicted numbers of events within each time interval match.

To set this up formally, recall that $F$ denotes the unknown true CDF. For each individual $x$, let $F_\theta(\mathbf{t} \mid x)$ denote the modeled CDF of time-until-failure. To measure overall calibration error, D-CALIBRATION accumulates the squared errors of the equality condition in Equation (2) over sets $I \in \mathcal{I}$ that cover $[0, 1]$:

$$\mathcal{R}(\theta) := \sum_{I \in \mathcal{I}} \left( \underset{P(\mathbf{t}, \mathbf{x})}{\mathbb{E}} \, \mathbb{1}\left[F_\theta(\mathbf{t} \mid \mathbf{x}) \in I\right] - |I| \right)^2. \tag{3}$$

The collection $\mathcal{I}$ is chosen to contain disjoint contiguous intervals $I \subseteq [0, 1]$, that cover the whole interval $[0, 1]$. Haider et al. [2020] perform a $\chi^2$-test to determine whether a model is well-calibrated, replacing the expectation in Equation (3) with a Monte Carlo estimate.

**Properties**  Setting aside the hypothesis testing step, we highlight two key properties of D-CALIBRATION.  First, D-CALIBRATION is zero for the correct conditional model.  This ensures that the correct model is not wrongly mischaracterized as miscalibrated. Second, for a given model class and dataset, smaller D-CALIBRATION means a model is more calibrated. This means that it makes sense to minimize D-CALIBRATION. Next, we make use of these properties and turn D-CALIBRATION into a differentiable objective.

## 3   X-CAL: A Differentiable Calibration Objective

We measure calibration error with D-CALIBRATION (Equation (3)) and propose to incorporate it into our training and minimize it directly. However, the indicator function $\mathbb{1}\left[\cdot\right]$ poses a challenge for optimization. Instead, we derive a soft version of D-CALIBRATION using a soft set membership function. We then develop an upper-bound to soft D-CALIBRATION that we call X-CAL that supports subsampling for stochastic optimization with batch data.

### 3.1   Soft Membership D-CALIBRATION

We replace the membership indicator for a set $I$ with a differentiable function. Let $\gamma > 0$ be a temperature parameter. Let $\sigma(x) = (1 + \exp[-x])^{-1}$. For point $u$ and the set $I = [a, b]$, define soft

membership $\zeta_\gamma$ as

$$\zeta_\gamma(u; I) := \sigma(\gamma(u - a)(b - u)), \tag{4}$$

where $\gamma \to \infty$ makes membership exact. This is visualized in Figure 2 in Appendix G. We propose the following differentiable approximation to Equation (3), which we call soft D-CALIBRATION, for use in a calibration objective:

$$\hat{\mathcal{R}}_\gamma(\theta) := \sum_{I \in \mathcal{I}} \left( \underset{P(\mathbf{t}, \mathbf{x})}{\mathbb{E}} \zeta_\gamma \left( F_\theta(\mathbf{t} \mid \mathbf{x}); I \right) - |I| \right)^2. \tag{5}$$

We find that $\gamma = 10^4$ allows for close-enough approximation to optimize exact D-CALIBRATION.

### 3.2 Stochastic Optimization via Jensen's Inequality

Soft D-CALIBRATION squares an expectation over the data, meaning that its gradient includes a product of two expectations over the same data. Due to this, it is hard to obtain a low-variance, unbiased gradient estimate with batches of data, which is important for models that rely on stochastic optimization. To remedy this, we develop an upper-bound on soft D-CALIBRATION, which we call X-CAL, whose gradient has an easier unbiased estimator.

Let $R_{\gamma,\theta}(t, x, I)$ denote the contribution to soft D-CALIBRATION error due to one set $I$ and a single sample $(t, x)$ in Equation (5): $R_{\gamma,\theta}(t, x, I) := \zeta_\gamma \left( F_\theta(t \mid x); I \right) - |I|$. Then soft D-CALIBRATION can be written as:

$$\hat{\mathcal{R}}_\gamma(\theta) = \sum_{I \in \mathcal{I}} \left( \underset{P(\mathbf{t}, \mathbf{x})}{\mathbb{E}} R_{\gamma,\theta}(\mathbf{t}, \mathbf{x}, I) \right)^2.$$

For each term in the sum over sets $I$, we proceed by in two steps. First, replace the expectation over data $\mathbb{E}_P$ with an expectation over sets of samples $\mathbb{E}_{S \sim P^M}$ of the mean of $R_{\gamma,\theta}$ where $S$ is a set of size $M$. Second, use Jensen's inequality to switch the expectation and square.

$$\hat{\mathcal{R}}_\gamma(\theta) = \sum_{I \in \mathcal{I}} \left( \underset{S \sim P^M}{\mathbb{E}} \frac{1}{M} \sum_{t,x \in S} R_{\gamma,\theta}(t, x, I) \right)^2 \leq \underset{S \sim P^M}{\mathbb{E}} \sum_{I \in \mathcal{I}} \left( \frac{1}{M} \sum_{t,x \in S} R_{\gamma,\theta}(t, x, I) \right)^2. \tag{6}$$

We call this upper-bound X-CAL and denote it by $\hat{\mathcal{R}}_\gamma^+(\theta)$. To summarize, $\lim_{\gamma \to \infty} \hat{\mathcal{R}}_\gamma(\theta) = \mathcal{R}(\theta)$ by soft indicator approximation and $\hat{\mathcal{R}}_\gamma(\theta) \leq \hat{\mathcal{R}}_\gamma^+(\theta)$ by Jensen's inequality. As $M \to \infty$, the slack introduced due to Jensen's inequality vanishes (in practice we are constrained by the size of the dataset). We now derive the gradient with respect to $\theta$, using $\zeta'(u) = \frac{d\zeta}{du}(u)$:

$$\frac{d\hat{\mathcal{R}}_\gamma^+}{d\theta} = \underset{S \sim P^M}{\mathbb{E}} \sum_{I \in \mathcal{I}} \frac{2}{M^2} \sum_{t,x \in S} R_{\gamma,\theta}(t, x, I) \left( \zeta'_\gamma \left( F_\theta(t \mid x); I \right) \frac{dF_\theta}{d\theta}(t \mid x) \right). \tag{7}$$

We estimate Equation (7) by sampling batches $S$ of size $M$ from the empirical data.

Analyzing this gradient demonstrates how X-CAL works. If the fraction of points in bin $I$ is larger than $|I|$, X-CAL pushes points out of $I$. The gradient of $\zeta_\gamma$ pushes points in the first half of the bin to have smaller CDF values and similarly points in the second half are pushed upwards.

While this works well for intervals not at the boundary of $[0, 1]$, some care must be taken at the boundaries. CDF values in the last bin may be pushed to one and unable to leave the bin. Since the maximum CDF value is one, $\mathbb{1}\left[u \in [a, 1]\right] = \mathbb{1}\left[u \in [a, b]\right]$ for any $b > 1$. Making use of this property, X-CAL extends the right endpoint of the last bin so that all CDF values are in the first half of the bin and therefore are pushed to be smaller. The boundary condition near zero is similar. We provide further analysis in Appendix I.

X-CAL can be added to loss functions such as negative log likelihood (NLL) and other survival modeling objectives such as Survival-CRPS (CRPS) [Avati et al., 2019]. For example, the full X-CALIBRATED maximum likelihood objective for a model $P_\theta$ and $\lambda > 0$ is:

$$\min_\theta \underset{P(\mathbf{t}, \mathbf{x})}{\mathbb{E}} - \log P_\theta(\mathbf{t} \mid \mathbf{x}) + \lambda \hat{\mathcal{R}}_\gamma^+(\theta). \tag{8}$$

**Choosing $\gamma$**  For small $\gamma$, soft D-CALIBRATION is a poor approximation to D-CALIBRATION. For large $\gamma$, gradients vanish, making it hard to optimize D-CALIBRATION. We find that setting $\gamma = 10000$ worked in all experiments. We evaluate the choice of $\gamma$ in Appendix G.

**Bound Tightness**  The slack in Jensen's inequality does not adversely affect our experiments in practice. We successfully use small batches, e.g. $< 1000$, for datasets such as MNIST. We always report exact D-CALIBRATION in the results. We evaluate the tightness of this bound and show that models ordered by the upper-bound are ordered in D-CALIBRATION the same way in Appendix H.

### 3.3  Handling Censored Data

In presence of right-censoring, failure times are censored more often than earlier times. So, applying the true CDF to only uncensored failure times results in a non-uniform distribution skewed to smaller values in $[0, 1]$. Censoring must be taken into account.

Let $x$ be a censored point with observed censoring time $u$ and unobserved failure time $\mathbf{t}$. Recall that $\boldsymbol{\delta} = \mathbb{1}\left[\mathbf{t} < \mathbf{c}\right]$. In this case $\mathbf{c} = \mathbf{u} = u$ and $\boldsymbol{\delta} = 0$. Let $F_{\mathbf{t}} = F(\mathbf{t} \mid x)$, $F_{\mathbf{c}} = F(\mathbf{c} \mid x)$, and $F_{\mathbf{u}} = F(\mathbf{u} \mid x)$. We first state the fact that, under $\mathbf{t} \perp\!\!\!\perp \mathbf{c} \mid \mathbf{x}$, a datapoint observed to be censored at time $u$ has $F_{\mathbf{t}} \sim \text{Unif}(F_u, 1)$ for true CDF $F$ (proof in Appendix C). This means that we can compute the probabilty that $\mathbf{t}$ falls in each bin $I = [a, b]$:

$$\mathbb{P}(F_{\mathbf{t}} \in I \mid \delta = 0, u, x) = \frac{(b - F_u)\mathbb{1}\left[F_u \in I\right]}{1 - F_u} + \frac{(b - a)\mathbb{1}\left[F_u < a\right]}{1 - F_u}, \tag{9}$$

Haider et al. [2020] make this observation and suggest a method for handling censoring points: they contribute $\mathbb{P}(F_{\mathbf{t}} \in I \mid \delta = 0, u, x)$ in place of the unobserved $\mathbb{1}\left[F_{\mathbf{t}} \in I\right]$:

$$\sum_{I \in \mathcal{I}} \left( \mathop{\mathbb{E}}_{\mathbf{u}, \boldsymbol{\delta}, \mathbf{x}} \left[ \boldsymbol{\delta} \mathbb{1}\left[F_{\mathbf{u}} \in I\right] + (1 - \boldsymbol{\delta})\mathbb{P}(F_{\mathbf{t}} \in I \mid \boldsymbol{\delta}, \mathbf{u}, \mathbf{x}) \right] - |I| \right)^2. \tag{10}$$

This estimator does not change the expectation defining D-CALIBRATION, thereby preserving the property that D-CALIBRATION is 0 for a calibrated model. We soften Equation (9) with:

$$\zeta_{\gamma, cens}(F_u; I) := \frac{(b - F_u)\sigma(\gamma(F_u - a)(b - F_u))}{(1 - F_u)} + \frac{(b - a)\sigma(\gamma(a - F_u))}{(1 - F_u)},$$

where we have used a one-sided soft indicator for $\mathbb{1}\left[F_u < a\right]$ in the right-hand term. We use $\zeta_{\gamma, cens}$ in place of $\zeta_{\gamma}$ for censored points in soft D-CALIBRATION. This gives the following estimator for soft D-CALIBRATION with censoring:

$$\sum_{I \in \mathcal{I}} \left( \mathop{\mathbb{E}}_{\mathbf{u}, \boldsymbol{\delta}, \mathbf{x}} \left[ \boldsymbol{\delta} \zeta_{\gamma}(F_\theta(\mathbf{u} \mid \mathbf{x}); I) + (1 - \boldsymbol{\delta})\zeta_{\gamma, cens}(F_\theta(\mathbf{u} \mid \mathbf{x}); I) \right] - |I| \right)^2. \tag{11}$$

The upper-bound of Equation (11) and its corresponding gradient can be derived analogously to the uncensored case. We use these in ours experiments on censored data.

## 4  Experiments

We study how X-CAL allows the modeler to optimize for a specified balance between prediction and calibration. We augment maximum likelihood estimation with X-CAL for various settings of coefficient $\lambda$, where $\lambda = 0$ corresponds to vanilla maximum likelihood. Maximum likelihood for survival analysis is described in Appendix A (Equation (12)). For the log-normal experiments, we also use Survival-CRPS (CRPS) [Avati et al., 2019] with X-CAL since S-CRPS enjoys a closed-form for log-normal. S-CRPS was developed to produce calibrated survival models but it optimizes neither a calibration measure nor a traditional likelihood. See Appendix B for a description of S-CRPS.

**Models, Optimization, and Evaluation**  We use log-normal, Weibull, Categorical and Multi-Task Logistic Regression (MTLR) models with various linear or deep parameterizations. For the discrete models, we optionally interpolate their CDF (denoted in the tables by NI for not-interpolated and I for interpolated). See Appendix E for general model descriptions. Experiment-specific model details may be found in Appendix F. We use $\gamma = 10000$. We use 20 D-CALIBRATION bins disjoint over

$[0, 1]$ for all experiments except for the cancer data, where we use 10 bins as in Haider et al. [2020]. For all experiments, we measure the loss on a validation set at each training epoch to chose a model to report test set metrics with. We report the test set NLL, test set D-CALIBRATION and Harrell's Concordance Index [Harrell Jr et al., 1996] (abbreviated CONC) on the test set for several settings of $\lambda$. We compute concordance using the Lifelines package [Davidson-Pilon et al., 2017]. All reported results are an average of three seeds.

**Data**   We discuss differences in performance on simulated gamma data, semi-synthetic survival data where times are conditional on the MNIST classes, length of stay prediction in the Medical Information Mart for Intensive Care (MIMIC-III) dataset [Johnson et al., 2016], and glioma brain cancer data from The Cancer Genome Atlas (TCGA). Additional data details may be found in Appendix D.

### 4.1   Experiment 1: Simulated Gamma Times with Log-Linear Mean

**Data**   We design a simulation study to show that a conditional distribution may achieve good concordance and likelihood but will have poor D-CALIBRATION. After adding X-CAL, we are able to improve the exact D-CALIBRATION. We sample $\mathbf{x} \in \mathbb{R}^{32}$ from a multivariate normal with $\sigma^2 = 10.0$. We sample times $\mathbf{t}$ conditionally from a gamma with mean $\boldsymbol{\mu}$ that is log-linear in $\mathbf{x}$ and constant variance 1e-3. The censoring times $\mathbf{c}$ are drawn like the event times, except with a different coefficient for the log-linear function. We experiment with censored and uncensored simulations, where we discard $\mathbf{c}$ and always observe $\mathbf{t}$ for uncensored. We sample a train/validation/test sets with 100k/50k/50k datapoints, respectively.

**Results**   Due to high variance in $\mathbf{x}$ and low conditional variance, this simulation has low noise. With large, clean data, this experiment validates the basic method on continuous and discrete models in the presence of censoring. Table 1 demonstrates how increasing $\lambda$ gracefully balances D-CALIBRATION with NLL and concordance for different models and objectives: log-normal trained via NLL and with S-CRPS, and the categorical model trained via NLL, without CDF interpolation. For results on more models and choices of $\lambda$ see Table 9 for uncensored results and Table 10 for censored in Appendix J.

**Table 1:**   Gamma simulation, censored

| | $\lambda$ | 0 | 1 | 10 | 100 | 500 | 1000 |
|---|---|---|---|---|---|---|---|
| Log-Norm | NLL | -0.059 | -0.049 | 0.004 | 0.138 | 0.191 | 0.215 |
| NLL | D-CAL | 0.029 | 0.020 | 0.005 | 2e-4 | 6e-5 | 7e-5 |
| | CONC | 0.981 | 0.969 | 0.942 | 0.916 | 0.914 | 0.897 |
| Log-Norm | NLL | 0.038 | 0.084 | 0.143 | 0.201 | 0.343 | 0.436 |
| S-CRPS | D-CAL | 0.017 | 0.007 | 0.001 | 1e-4 | 5e-5 | 8e-5 |
| | CONC | 0.982 | 0.978 | 0.963 | 0.950 | 0.850 | 0.855 |
| Cat-NI | NLL | 0.797 | 0.799 | 0.822 | 1.149 | 1.665 | 1.920 |
| | D-CAL | 0.009 | 0.006 | 0.002 | 2e-4 | 6e-5 | 6e-5 |
| | CONC | 0.987 | 0.987 | 0.987 | 0.976 | 0.922 | 0.861 |

### 4.2   Experiment 2: Semi-Synthetic Experiment: Survival MNIST

**Data**   Following Pölsterl [2019], we simulate a survival dataset conditionally on the MNIST dataset [LeCun et al., 2010]. Each MNIST label gets a deterministic risk score, with labels loosely grouped together by risk groups (Table 5 in Appendix D.2). Datapoint image $\mathbf{x}_i$ with label $\mathbf{y}_i$ has time $\mathbf{t}_i$ drawn from a Gamma with mean equal to $\text{risk}(y_i)$ and constant variance 1e-3. Therefore $\mathbf{t}_i \perp\!\!\!\perp \mathbf{x}_i \mid \mathbf{y}_i$ and times for datapoints that share an MNIST class are identically drawn. We draw censoring times $\mathbf{c}$ uniformly between the minimum failure time and the $90^{th}$ percentile time, which resulted in about 50% censoring. We use PyTorch's MNIST with test split into validation/test. The model does not see the MNIST class and learns a distribution over times given pixels $\mathbf{x}_i$. We experiment with censored and uncensored simulations, where we discard $\mathbf{c}$ and always observe $\mathbf{t}$ for uncensored.

**Results**   This semi-synthetic experiment tests the ability to tune calibration in presence of a high-dimensional conditioning set (MNIST images) and through a typical convolutional architecture. Table 2

demonstrates that the deep log-normal models started off miscalibrated relative to the categorical model for $\lambda = 0$ and that all models were able to significantly improve in calibration. See Table 11 and Table 12 for more uncensored and censored survival-MNIST results.

**Table 2:** Survival-MNIST, censored

| | $\lambda$ | 0 | 1 | 10 | 100 | 500 | 1000 |
|---|---|---|---|---|---|---|---|
| Log-Norm | NLL | 4.337 | 4.377 | 4.483 | 4.682 | 4.914 | 5.151 |
| NLL | D-CAL | 0.392 | 0.074 | 0.020 | 0.005 | 0.005 | 0.007 |
| | CONC | 0.902 | 0.873 | 0.794 | 0.696 | 0.628 | 0.573 |
| Log-Norm | NLL | 4.950 | 4.929 | 4.859 | 4.749 | 4.786 | 4.877 |
| S-CRPS | D-CAL | 0.215 | 0.122 | 0.051 | 0.010 | 0.002 | 9e-4 |
| | CONC | 0.891 | 0.881 | 0.874 | 0.868 | 0.839 | 0.815 |
| Cat-NI | NLL | 1.733 | 1.734 | 1.765 | 1.861 | 2.074 | 3.030 |
| | D-CAL | 0.018 | 0.014 | 0.004 | 5e-4 | 5e-4 | 4e-4 |
| | CONC | 0.945 | 0.945 | 0.927 | 0.919 | 0.862 | 0.713 |

## 4.3  Experiment 3: Length of Stay Prediction in MIMIC-III

**Data**   We predict the length of stay (in number of hours) in the ICU, using data from the MIMIC-III dataset. Such predictions are important both for individual risk predictions and prognoses and for hospital-wide resource management. We follow the preprocessing in Harutyunyan et al. [2017], a popular MIMIC-III benchmarking paper and repository [3]. The covariates are a time series of 17 physiological variables (Table 6 in Appendix D.3) including respiratory rate and glascow coma scale information. There is no censoring in this task. We skip imputation and instead use missingness masks as features. There are $2,925,434$ and $525,912$ instances in the training and test sets. We split the training set in half for train and validation.

**Results**   Harutyunyan et al. [2017] discuss the difficult of this task when predicting fine-grained lengths-of-stay, as opposed to simpler classification tasks like more/less one week stay. The true conditionals are high in entropy given the chosen covariates Table 3 demonstrates this difficulty, as can be seen in the concordances. We report the categorical model with and without CDF interpolation and the log-normal trained with S-CRPS. NLL for the log-normal is not reported because S-CRPS does not optimize NLL and did poorly on this metric. The log-normal trained with NLL was not able to fit this task on any of the three metrics. All three models reported are able to reduce D-CALIBRATION. Results for all models and more choices of $\lambda$ may be found in Table 13. The categorical models with and without CDF interpolation match in concordance for $\lambda = 0$ and $\lambda = 1000$. However, the interpolated model achieves better D-CALIBRATION. This may be due to the lower-bound $\ell > 0$ on a discrete model's D-CALIBRATION (Appendix E).

**Table 3:** MIMIC-III length of stay

| | $\lambda$ | 0 | 1 | 10 | 100 | 500 | 1000 |
|---|---|---|---|---|---|---|---|
| Log-Norm | D-CAL | 0.859 | 0.639 | 0.155 | 0.046 | 0.009 | 0.005 |
| S-CRPS | CONC | 0.625 | 0.639 | 0.575 | 0.555 | 0.528 | 0.506 |
| Cat-NI | Test NLL | 3.142 | 3.177 | 3.167 | 3.088 | 3.448 | 3.665 |
| | D-CAL | 0.002 | 0.002 | 0.001 | 2e-4 | 1e-4 | 1e-4 |
| | CONC | 0.702 | 0.700 | 0.699 | 0.690 | 0.642 | 0.627 |
| Cat-I | NLL | 3.142 | 3.075 | 3.073 | 3.073 | 3.364 | 3.708 |
| | D-CAL | 4e-4 | 2e-4 | 2e-4 | 1e-4 | 5e-5 | 4e-5 |
| | CONC | 0.702 | 0.702 | 0.702 | 0.695 | 0.638 | 0.627 |

## 4.4 Experiment 4: Glioma data from The Cancer Genome Atlas

We use the glioma (a type of brain cancer) dataset [4] collected as part of the TCGA program and studied in [Network, 2015]. We focus on predicting time until death from the clinical data, which includes tumor tissue location, time of pathological diagnosis, Karnofsky performance score, radiation therapy, demographic information, and more. Censoring means they did not pass away. The train/validation/test sets are made of 552/276/277 datapoints respectively, of which 235/129/126 are censored, respectively.

**Results** For this task, we study the Weibull AFT model, reduce the deep log-normal model from three to two hidden layers, and study a linear MTLR model (with CDF interpolation) in place of the deep categorical due to the small data size. MTLR is more constrained than linear categorical due to shared parameters. Table 4 demonstrates these three models' ability to improve D-CALIBRATION. MTLR is able to fit well and does not give up much concordance. Results for all models and more choices of $\lambda$ may be found in Table 14.

**Table 4:** The Cancer Genome Atlas, glioma

|            | $\lambda$ | 0      | 1     | 10    | 100   | 500   | 1000  |
|------------|-----------|--------|-------|-------|-------|-------|-------|
| Log-Norm   | NLL       | 14.187 | 6.585 | 4.639 | 4.181 | 4.403 | 4.510 |
| NLL        | D-CAL     | 0.059  | 0.024 | 0.010 | 0.003 | 0.002 | 0.004 |
|            | CONC      | 0.657  | 0.632 | 0.703 | 0.805 | 0.474 | 0.387 |
| Weibull    | NLL       | 4.436  | 4.390 | 4.292 | 4.498 | 4.475 | 4.528 |
|            | D-CAL     | 0.035  | 0.028 | 0.009 | 0.003 | 0.004 | 0.007 |
|            | CONC      | 0.788  | 0.785 | 0.777 | 0.702 | 0.608 | 0.575 |
| MTLR-NI    | NLL       | 1.624  | 1.620 | 1.636 | 1.658 | 1.748 | 1.758 |
|            | D-CAL     | 0.009  | 0.007 | 0.005 | 0.003 | 0.002 | 0.002 |
|            | CONC      | 0.828  | 0.829 | 0.824 | 0.818 | 0.788 | 0.763 |

## 5 Related Work

**Deep Survival Analysis** Recent approaches to survival analysis parameterize the failure distribution as a deep neural network function of the [Ranganath et al., 2016, Alaa and van der Schaar, 2017, Katzman et al., 2018]. Miscouridou et al. [2018] and Lee et al. [2018] use a discrete categorical distribution over times interpreted ordinally, which can approximate any smooth density with sufficient data. The categorical approach has also been used when the conditional is parameterized by a recurrent neural network of sequential covariates [Giunchiglia et al., 2018, Ren et al., 2019]. Miscouridou et al. [2018] extend deep survival analysis to deal with missingness in x.

**Post-training calibration methods** Practitioners have used two calibration methods for binary classifiers, which modify model predictions maximize likelihood on a held-out dataset. Platt scaling [Platt, 1999] works by using a scalar logistic regression built on top of predicted probabilities. Isotonic regression [Zadrozny and Elkan, 2002] uses a nonparametric piecewise linear transformation instead of the logistic regression. These methods do not reveal an explicit balance between prediction quality and calibration during model training. X-CAL allows practitioners to explore this balance while searching in the full model space.

**Objectives** When an unbounded loss function (e.g. NLL) is used and the gradients are a function of $x$, the model may put undue focus on explaining a given outlier $x^\star$, worsening calibration during training. For this reason, robust objectives have been explored. Avati et al. [2019] consider continuous ranked probability score (CRPS) [Matheson and Winkler, 1976], a robust proper scoring rule for continuous outcomes, and adapt it to S-CRPS for survival analysis by accounting for censoring. However, S-CRPS does not provide a clear way to balance predictive power and calibration. Kumar

et al. [2018] develop a trainable kernel-based calibration measure for binary classification but they do not discuss an optimizable calibration metric for survival analysis.

**Brier Score**    The Brier Score [Brier and Allen, 1951] decomposes into a calibration metric (numerator of Hosmer-Lemeshow) and a discrimination term encouraging patients with the same failure status at $t^\star$ to have the same failure probability at $t^\star$. To capture entire distributions over time, the Integrated Brier Score is used. The Inverse Probability of Censoring Weighting Brier Score [Graf et al., 1999] handles censoring but requires estimation of the censoring distribution, a whole survival analysis problem (with censoring due to the failures) on its own [Gerds and Schumacher, 2006, Kvamme and Borgan, 2019]. X-CAL can balance discrimination and calibration without estimation of the censoring distribution.

## 6    Discussion

Model calibration is an important consideration in many clinical problems, especially when treatment decisions require risk estimates across all times in the future. We tackle the problem of building models that are calibrated over individual failure distributions. To this end, we provide a new technique that explicitly targets calibration during model training. We achieve this by constructing a differentiable approximation of D-CALIBRATION, and using it as an add-on objective to maximum likelihood and S-CRPS. As we show in our experiments, X-CAL allows for explicit and direct control of calibration on both simulated and real data. Further, we showed that searching over the X-CAL $\lambda$ parameter can strike the practitioner-specified balance between predictive power and calibration.

**Marginal versus Conditional Calibration**    D-CALIBRATION is $0$ for the true conditional and marginal distributions of failure times. This is because D-CALIBRATION measures marginal calibration, i.e. $\mathbf{x}$ is integrated out. Conditional calibration is the stronger condition that $F_\theta(t \mid x)$ is calibrated for all $x$. This is in general infeasible even to measure (let alone optimize) [Vovk et al., 2005, Pepe and Janes, 2013, Barber et al., 2019] without strong assumptions since for continuous $x$ we usually observe just one sample. However, among the distributions that have $0$ D-CALIBRATION, the true conditional distribution has the smallest NLL. Therefore, X-CALIBRATED objectives with proper scoring rules (like NLL) have an optimum only for the true conditional model in the limit of infinite data.

**D-Calibration and Censoring**    Equation (10) in Section 3.3 provides a censored version of D-CALIBRATION that is $0$ for a calibrated model, like the original D-CALIBRATION (Equation (3)). However, this censored calibration measure is not equal to D-CALIBRATION in general for miscalibrated models. For a distribution $F_\theta$ with non-zero D-CALIBRATION, for any censoring distribution $G$, estimates of the censored version will assess $F_\theta$ to be more uniform than if exact D-CALIBRATION were able to be computed using all true observed failure times. This happens especially in the case of heavy and early censoring because a lot of uniform weight is assigned [Haider et al., 2020, Avati et al., 2019]. This means that the censored objective can be close to $0$ for a miscalibrated model on a highly censored dataset.

An alternative strategy that avoids this issue is to use inverse weighting methods (e.g. Inverse Propensity Estimator of outcome under treatment [Horvitz and Thompson, 1952], Inverse Probability of Censoring-Weighted Brier Score [Graf et al., 1999, Gerds and Schumacher, 2006] and Inverse Probability of Censoring-Weighted binary calibration for survival analysis [Yadlowsky et al., 2019]). Inverse weighting would preserve the expectation that defines D-CALIBRATION for any censoring distribution. One option is to adjust with $p(\mathbf{c} \mid \mathbf{x})$. This requires $\mathbf{c} \perp \mathbf{t} \mid \mathbf{x}$ and solving an additional censored survival problem $p(\mathbf{c} \mid \mathbf{x})$. Nevertheless, if a censoring estimate is provided, the methodology in this work could then be applied to an inverse-weighted D-CALIBRATION. There is then a trade-off between the censored estimator proposed by Haider et al. [2020] that we use (no modeling $G$) and inverse-weighted estimators (which preserve D-CALIBRATION for miscalibrated models).

## Broader Impact

In this paper, we study calibration of survival analysis models and suggest an objective for improving calibration during model training. Since calibration means that modeled probabilities correspond to

the actual observed risk of an event, practitioners may feel more confident about using model outputs directly for decision making e.g. to decide how many emergency room staff members qualified for performing a given procedure should be present tomorrow given all current ER patients. But if the distribution of event times in these patients differs from validation data, because say the population has different demographics, calibration should not provide the practitioner with more confidence to directly use such model outputs.

## Acknowledgments

This work was supported by:

- NIH/NHLBI Award R01HL148248
- NSF Award 1922658 NRT-HDR: FUTURE Foundations, Translation, and Responsibility for Data Science.
- NSF Award 1514422 TWC: Medium: Scaling proof-based verifiable computation
- NSF Award 1815633 SHF

We thank Humza Haider for sharing the original D-calibration experimental data, Avati et al. [2019] for publishing their code and the Cancer Genome Atlas Research Network for making the glioma data public. We thank all the reviewers for thoughtful feedback.

## Footnotes

[2]Code is available at https://github.com/rajesh-lab/X-CAL

[3]https://github.com/YerevaNN/mimic3-benchmarks

[4]https://www.cancer.gov/about-nci/organization/ccg/research/structural-genomics/tcga/studied-cancers/glioma

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
