[Supplementary Material]

# A    Background on Survival Analysis and Related Work

Survival analysis models the probability distribution of a time-until-event. The event is often called a failure time. For example, we may model time until onset of coronary heart disease given a patient's current health status [Wilson et al., 1998, Vasan et al., 2008].

Survival analysis differs from standard probabilistic regression problems in that data may be censored. For example, a patient may leave a study before developing the studied condition, or may not develop the condition before the study ends. In these cases, the time that a patient leaves or the study ends is called the censoring time. These are cases of right-censoring, where it is only known that the failure time is greater than the observed censoring time.

We review key definitions in survival analysis. See George et al. [2014] for a review. For textbooks, see Andersen et al. [2012], Kalbfleisch and Prentice [2002], and Lawless [2011].

## A.1    Notation

Let $\mathbf{t}$ be a continuous random variable denoting the failure time with CDF $F$ and density $f$. The survival function $\overline{F}$ is defined as 1 minus the CDF: $\overline{F} = 1 - F$. Censoring times are considered random variables $\mathbf{c}$ with CDF $G$, survival function $\overline{G}$, and density $g$. In general these distributions may be conditional on covariates $\mathbf{x}$.

For datapoints $i$, let $\mathbf{t}_i$ be failure times and $\mathbf{c}_i$ be censoring times. Let us focus on right-censoring where $\mathbf{u}_i = \min(\mathbf{t}_i, \mathbf{c}_i)$, $\boldsymbol{\delta}_i = \mathbb{1}\left[\mathbf{t}_i < \mathbf{c}_i\right]$ and the observed data consists of $(x_i, u_i, \delta_i)$. In general we cannot throw away censored points, since $p(t \mid x, t < c) \neq p(t \mid x)$ and we would therefore biasedly estimate the failure distribution $F$.

## A.2    Assumptions About Censoring

It may seem that we need to model $\mathbf{c}$ to estimate the parameters of $f$, but under certain assumptions, we can write the likelihood (with respect to $f$'s parameters) for a dataset with censoring without estimating the censoring distribution. In this work, we assume:

**Assumption.** Censoring-at-random. $\mathbf{t}$ is distributed marginally or conditionally on $\mathbf{x}$. $\mathbf{c}$ is either a constant, distributed marginally, or distributed conditionally on $\mathbf{x}$. In any case, it must hold that $\mathbf{t} \perp\!\!\!\perp \mathbf{c} \mid \mathbf{x}$.

**Assumption.** Non-informative Censoring. The censoring time $\mathbf{c}$'s distribution parameters $\theta_c$ are distinct from parameters $\theta_t$ of $\mathbf{t}$'s distribution.

## A.3    Likelihoods

Under the two censoring assumptions, the log-likelihood can be derived to be

$$L(\theta_t) = \sum_i \delta_i \log f_{\theta_t}(t_i \mid x_i) + (1 - \delta_i) \log \overline{F}_{\theta_t}(t_i \mid x_i) \tag{12}$$

and can be maximized to learn parameters $\theta_t$ of $f$ without an estimate of $G$. This can be interpreted as follows: an uncensored individual has $\delta_i = 1$, meaning $u_i = t_i$. This point contributes through the failure density $f(u_i) = f(t_i)$, as in standard regression likelihoods. Censored points contribute through failure survival function $\overline{F} = 1 - F$ because there failure time is known to be greater than $u_i$. Full discussions of survival likelihoods can be found in Kalbfleisch and Prentice [2002], Lawless [2011], Andersen et al. [2012].

## A.4    Testing Calibration

Classical goodness-of-fit tests [Lemeshow and Hosmer Jr, 1982, Grønnesby and Borgan, 1996, D'agostino and Nam, 2003] and their recent modifications [Demler et al., 2015] assess calibration of survival analysis models for a particular time of interest $t^*$. These take the following steps:

1. pick a time $t^\star$ at which to measure calibration
2. evaluate model probability $p_i = p_\theta(\mathbf{t} < t^\star \mid \mathbf{x}_i)$ of failing by time $t^\star$

**Figure 1:** Sub-optimal calibration curves that result in optimal calibration slope.

3. sort $p_i$ into $K$ groups $g_k$ defined by quantiles (e.g. $K = 2$ corresponds to partitioning the data into a low-risk group and high-risk group)

4. compute the *observed* # of events using e.g. $(1 - \text{KM}_k[t^*])|g_k|$ where $\text{KM}_k$ the Kaplan-Meier estimate [Kaplan and Meier, 1958] of the survival function just on data in $g_k$'s

5. compute the *expected* #, $E_k = \sum_{i \in g_k} p_i$

6. let $\bar{p}_k = \frac{1}{|g_k|} \sum_{i \in g_k} p_i$

7. $\sum_k \frac{(O_k - E_k)^2}{|g_k|\bar{p}_k(1 - \bar{p}_k)}$ gives a $\chi^2$ test statistic

8. small p-value $\rightarrow$ model not calibrated

Demler et al. [2015] review these tests and propose some modifications when there are not enough individuals assigned to each bin. These tests are limited in two ways: they answer calibration in a rigid yes/no fashion with hypothesis testing, and it is not clear how to combine calibration assessments over the entire range of possible time predictions.

### A.5 Calibration Slope

**Calibration Slope**   Recent publications in machine learning [Avati et al., 2019] and in medicine [Besseling et al., 2017] use the *calibration slope* to evaluate calibration [Stevens and Poppe, 2020]. First, a calibration curve is computed by plotting, for each quantile $\rho \in [0, 1]$, the fraction of observed samples with a failure time smaller than that quantile's time $t(\rho) = F_\theta^{-1}(\rho \mid x)$. Then, report the slope of the best-fit line to this curve. When a model is well-calibrated, the true and predicted densities are close and the best fit line has slope 1.0. However, slope can be 1.0 (with intercept 0.0) even when the model is not well-calibrated.

Here, we construct two possible calibration curves that cannot result from well-calibrated models. However, the resulting calibration slope is close to 1.0. Avati et al. [2019] use a line of best fit with non-zero intercept. We plot hypothetical calibration curves in Figure 1 such that the corresponding best fit line has slope 1.0, with and without intercept terms. Stevens and Poppe [2020] make a related observation about calibration slope: a near-zero intercept of the line of best fit, or other evidence of calibration, should always be reported alongside near-1 slope when claiming a model is calibrated. However, we demonstrate here that even slope 1 and intercept 0 can result from poorly calibrated models. The interested reader should see Stevens and Poppe [2020] for an assessment of recent publications in medicine that report only slope and for the history of slope-only as a "measure of spread" [Cox, 1958].

## B   Survival CRPS

S-CRPS is proposed by Avati et al. [2019]:

$$\mathcal{S}_{\text{CRPS}}(\hat{F}, (y, c)) = \int_0^y \hat{F}(z)^2 dz + (1 - c) \int_y^\infty (1 - \hat{F}(z))^2 dz,$$

where $y$ is the event time, $c$ is an indicator for censorship and $\hat{F}$ is the CDF from the model. See Avati et al. [2019] Appendix B for a detailed derivation of S-CRPS objective for a log-normal model.

## C CDF of Survival Time is Uniform for Censored Patient

Consider the data distribution $P(\mathbf{t}, \mathbf{c} \mid x)$ and using the conditional $P(\mathbf{t} \mid x)$ of this distribution to evaluate D-CALIBRATION on this data. For a point that is censored at time $c$, $P(\mathbf{t} \mid x)$ would simply condition on the event $\mathbf{t} > c$ for constant $c$, yielding $P(\mathbf{t} \mid \mathbf{t} > c, x)$. However, the true failure distribution for such a point is $P(\mathbf{t} \mid \mathbf{t} > c, \mathbf{c} = c, x)$. Under censoring-at-random,

$$\mathbf{t} \perp\!\!\!\perp \mathbf{c} \mid \mathbf{x} \implies P(\mathbf{t} \mid \mathbf{t} > c, x) = P(\mathbf{t} \mid \mathbf{t} > c, \mathbf{c} = c, x). \tag{13}$$

Let $F$ be the failure CDF. Let $p_t$ be the density of $\mathbf{t} \mid x$. Apply transformation $\mathbf{z} = F(\mathbf{t}|x)$. To compute $\mathbf{z}$'s density, we need:

$$\frac{d}{dz} F^{-1}(z|x) = \frac{1}{p_t(F^{-1}(z \mid x))} = \frac{1}{p_t(t)}.$$

Applying change of variable to compute $\mathbf{z}$'s density:

$$p_t(F^{-1}(z|x)) \frac{d}{dz} F^{-1}(z|x) = p_t(t) \frac{1}{p_t(t)} = 1$$

Therefore, $\mathbf{z}$ is uniform distributed over [0,1]. So conditioning on set $(\mathbf{t} > c, x) = (\mathbf{z} > F(c|x), x)$ gives the result:

$$\mathbf{z} \mid (\mathbf{t} > c, x) \sim \text{Unif}(F(c \mid x), 1).$$

The CDF value of the unobserved time for a censored datapoint is uniform above the failure CDF applied to the censoring time. Haider et al. [2020] (Appendix B) give an alternate proof in terms of expected counts.

## D Extra Data Details

### D.1 Data Details for Simulation Study

For the gamma simulation, we draw $\mathbf{x}$ from a $D = 32$ multivariate Normal with $\mathbf{0}$ mean and diagonal covariance with $\sigma^2 = 10.0$. We draw failure times $\mathbf{t}$ conditionally on $\mathbf{x}$ from a gamma distribution with mean $\boldsymbol{\mu}$ log-linear in $\mathbf{x}$. The weights of the linear function are drawn uniformly. The gamma distribution has constant variance 1e-3. This is achieved by setting $\alpha = \boldsymbol{\mu}_i^2/1e\text{-}3$ and $\beta = \boldsymbol{\mu}_i/1e\text{-}3$.

$$\mathbf{x}_i \sim \mathcal{N}(0, \sigma^2 \mathbf{I}), \quad \mathbf{w}_d \sim \text{Unif}(-0.1, 0.1), \quad \boldsymbol{\mu}_i = \exp[\mathbf{w}^\top \mathbf{x}_i], \quad \mathbf{t}_i \sim \text{Gamma}(\alpha, \beta).$$

Censoring times are drawn like failure times but with a different set of weights for the linear function. This means $\mathbf{t} \perp\!\!\!\perp \mathbf{c} \mid \mathbf{x}$.

### D.2 Data Details for MNIST

As described in the main text, we follow Pölsterl [2019] to simulate a survival dataset conditionally on the MNIST dataset [LeCun et al., 2010]. Each MNIST label gets a deterministic risk score, with labels loosely grouped together by risk groups. See Table 5 for an example of the risk groups and risk scores for the MNIST classes.

Datapoint image $\mathbf{x}_i$ with label $\mathbf{y}_i$ has time $\mathbf{t}_i$ drawn from a Gamma whose mean is the risk score and whose variance is constant 1e-3. Therefore $\mathbf{t}_i$ is independent of $\mathbf{x}_i$ given $\mathbf{y}_i$ and times for datapoints that share an MNIST class are identically drawn.

$$\boldsymbol{\mu}_i = \text{risk}(\mathbf{y}_i) \quad v = 1e\text{-}3 \quad \alpha = \boldsymbol{\mu}_i^2/v, \quad \beta = \boldsymbol{\mu}_i/v, \quad \mathbf{t}_i \sim \text{Gamma}(\alpha, \beta)$$

For each split of the data (e.g. training set), we draw censoring times uniformly between the minimum failure time in that split and the $90^{th}$ percentile time, which, due to the particular failure distributions, resulted in about 50% censoring.

**Table 5:** Risk scores for digit classes.

| Digit | 0 | 1 | 2 | 3 | 4 | 5 | 6 | 7 | 8 | 9 |
|---|---|---|---|---|---|---|---|---|---|---|
| Risk Group | most | least | lower | lower | lower | higher | least | most | least | most |
| Risk Score | 11.25 | 2.25 | 5.25 | 5.0 | 4.75 | 8.0 | 2.0 | 11.0 | 1.75 | 10.75 |

**Table 6:** The 17 selected clinical variables. The second column shows the source table(s) of a variable from MIMIC-III database. The third column lists the "normal" values used in the imputation step. Table reproduced from Harutyunyan et al. [2017].

| Variable table | Impute value | Modeled as |
|---|---|---|
| Capillary refill rate | 0.0 | categorical |
| Diastolic blood pressure | 59.0 | continuous |
| Fraction inspired oxygen | 0.21 | continuous |
| Glasgow coma scale eye opening | 4 spontaneously | categorical |
| Glasgow coma scale motor response | 6 obeys commands | categorical |
| Glasgow coma scale total | 15 | categorical |
| Glasgow coma scale verbal response | 5 oriented | categorical |
| Glucose | 128.0 | continuous |
| Heart Rate | 86 | continuous |
| Height | 170.0 | continuous |
| Mean blood pressure | 77.0 | continuous |
| Oxygen saturation | 98.0 | continuous |
| Respiratory rate | 19 | continuous |
| Systolic blood pressure | 118.0 | continuous |
| Temperature | 36.6 | continuous |
| Weight | 81.0 | continuous |
| pH | 7.4 | continuous |

### D.3 Data Details for MIMIC-III

We show the 17 physiological variables we use in Table 6. The table is reproduced from Harutyunyan et al. [2017]. This dataset differs from other MIMIC-III length of stay datasets because one stay in the ICU of a single patient produces many datapoints: remaining time at each hour after admission. After excluding ICU transfers and patients under 18, there are $2,925,434$ and $525,912$ instances in the training and test sets. We split the training set in half for train and validation.

### D.4 Data Details for The Cancer Genome Atlas Glioma Data

We use the glioma (a type of brain cancer) data[5] collected as part of the TCGA program and studied in [Network, 2015]. TCGA comprises clinical data and molecular from 11,000 patients being treated for a diverse set of cancer types. We focus on predicting time until death from the clinical data, which includes:

- tumor tissue site
- time of initial pathologic diagnosis
- radiation therapy
- Karnofsky performance score
- histological type
- demographic information

Censoring means they did not pass away. The train/validation/test sets are made of 552/276/277 datapoints respectively, of which 235/129/126 are censored, respectively.

To download this data, use the firebrowse. tool, select the Glioma (GBMLGG) cohort, and then click the blue clinical features bar on the right hand side. Select the "Clinical Pick Tier 1" file.

We standardized the features and then clamped their maximum absolute value at $5.0$. This is in part because we were working with the Weibull AFT model, which is very sensitive to large variance in covariates.

# E  Model Descriptions

We describe the models we use in the experiments. For all models, the parameterization as a function of $\mathbf{x}$ varies in complexity (e.g. linear or deep) depending on task.

**Log-normal model**  When $\log T$ is Normal with mean $\mu$ and variance $\sigma^2$, we say that $T$ is log-normal with location $\mu$ and scale $\sigma$. We parameterize $\mu$ and $\sigma$ as functions of $\mathbf{x}$ (small ReLU networks with 1 to 3 hidden layers, depending on experiment).

**Weibull Model**  The Weibull Accelerated Failure Times (AFT) model sets $\log T = \beta_0 + \beta^\top X + \sigma W$ where $\sigma$ is a scale parameter and $W$ is Gumbel. It follows that $T \sim \text{Weibull}(\lambda, k)$ with scale $\lambda = \exp[\beta^\top X]$ and concentration $k = \sigma^{-1}$ [Liu, 2018]. We constrain $k \in (1, 2)$.

**Interpolation for Discrete Models**  The next two models predict for a finite set of times and therefore have a discontiuous CDF. These models have a lower-bound $\ell > 0$ on D-CALIBRATION because the CDF values will not be $\text{Unif}(0, 1)$ distributed. However, $\ell$ decreases to $0$ as the number of discrete times increases. For any fixed number of times, minimizing D-CALIBRATION will still improve calibration, which we observe in our experiments.

We optionally use linear interpolation to calculate the CDF. Suppose a time $t$ falls into bin $k$ which covers time interval $(t_a, t_b)$. If we do not use interpolation, then the CDF value $P(T \le t)$ we calculate is the sum of the probabilities of bins whose indices are smaller than or equal to $k$. If we use linear interpolation, we replace the probability of bin $k$, $P(k)$, in the summation by:

$$\frac{t - t_a}{t_b - t_a} P(k)$$

**Categorical Model**  We parameterize a categorical distribution over discrete times by using a neural network function of $\mathbf{x}$ with a size $B$ output. Interpreted ordinally, this can approximate continuous survival distributions as $B \to \infty$ [Lee et al., 2018, Miscouridou et al., 2018]. The time for each bin is set to training data percentiles so that each next bin captures the range of times for the next $(100/B)^{th}$ percentile of training data, using only uncensored times.

**Multi-Task Logistic Regression (mtlr)**  MTLR differs from the Categorical Model because there is some relationship between the probability of the bins. Assume we have $K$ bins. In the linear case Yu et al. [2011], suppose our input is $x$ and parameters $\Theta = (\theta_1, \ldots, \theta_{K-1})$. The probability for bin $k < K$ is:

$$\frac{\exp(\sum_{j=k}^{K-1} \theta_j^T x)}{1 + \sum_{i=1}^{K-1} \exp(\sum_{j=i}^{K-1} \theta_j^T x)},$$

and the probability for bin $K$ is :

$$\frac{1}{1 + \sum_{i=1}^{K-1} \exp(\sum_{j=i}^{K-1} \theta_j^T x)}.$$

# F  Experimental Details

## F.1  Gamma Simulation

We use a 4-layer neural network of hidden-layer sizes $128, 64, 64$ units, with ReLU activations to parameterize the categorical and log-normal distributions. For categorical we use another linear

transformation to map to 50 output dimensions. For the log-normal model, two copies of the above neural network are used, one to output the location and the other to output the log of the log-normal scale parameter. For MTLR, we use a linear transformation from covariates to 50 dimensions and use a softmax layer to output the probability for the 50 bins. We use $0$ dropout, $0$ weight decay, learning rate 1e-3 and batch size 1000 for 100 epochs in this experiment.

### F.2 Survial MNIST

The model does not see the MNIST class and learns a distribution over times given pixels $\mathbf{x}_i$. We use a convolutional neural network. We use several layers of 2D convolutions with a kernel of size 2 and stride of size 1. The sequence of channel numbers is $32, 64, 128, 256$ with the last layer containing scalars. After each convolution, we use ReLU, then dropout, then size 2 max pooling.

For categorical and log-normal models, this CNN output is mapped through a three-hidden-layer ReLU neural network with hidden sizes $512, 1024, 1024$. Between the fully connected layers, we use ReLU then dropout. Again, with the log-normal, separate networks are used to output the location and log-scale. For MTLR, the CNN output is linearly mapped to the 50 bins. For categorical, we use $0.2$ dropout for uncensored and $0.1$ for censored. In MTLR, we use dropout $0.2$. In lognormal, we use dropout $0.1$. We use weight decay 1e-4, learning rate 1e-3, and batch size 5000 for 200 epochs.

### F.3 MIMIC-III

The input is high-dimensional (about $1400$) because it is a concatenated time series and because missingness masks are used. We use a 4-layer neural network of hidden-layer sizes $2048, 1024, 1024$ units with ReLU activations. For the categorical model, we use $B = 20$ categorical output bins. For the log-normal model, we use one three-hidden neural network of hidden-layer sizes $128, 64, 64$ units and an independent copy to output the location and log-scale parameters. We use dropout $0.15$, learning rate 1e-3 and weight decay 1e-4 for 200 epochs at batch size 5000.

### F.4 The Cancer Genome Atlas, Glioma

The Weibull model has parameters scale and concentration. The scale is set to $\exp[\beta^\top \mathbf{x}]$ for regression parameters $\beta$, plus a constant $1.0$ for numerical stability. We optimize the concentration parameter in $(1, 2)$. The log-normal model is as described in the simulated gamma experiment, except that it has two instead of three hidden layers, due to small data sample size. The categorical and MTLR models are also as described in the simulated gamma experiment, except that they have 20 instead of 50 bins, and are linear, again due to small data sample size.

We standardize this data and then clamp all covariates at absolute value $5.0$. For all models, we train for 10,000 epochs at learning rate 1e-2 with full data batch size 1201. We use 10 D-CALIBRATION bins for this experiment as studied in Haider et al. [2020], rather than the 20 bins used in all other experiments.

## G Exploring Choice of $\gamma$ soft-indicator parameter

There is a trade-off in setting the soft membership parameter $\gamma$. Larger values approximate the indicator function better, but can have worse gradients because the values lie in the flat region of the sigmoid. See Figure 2 for an example of how gamma changes the soft indicator for a given set $I = [0.45, 0.55]$. We choose $\gamma = 10000$ in all of the experiments and find that it allows us to minimize exact d-cal (D-CAL). We explore other choices in Table 7. We see the expected improvement in approximation as $\gamma$ increases. Then, as $\gamma$ gets too large, exact D-CAL stops improving as a function of $\lambda$.

## H Exploring Slack due to Jensen's Inequality

We trained the Categorical model on the gamma simulation data with $\gamma = 10,000$ and batch size $10,000$ for all $\lambda$. The trained models are evaluated on the training set (size $100,000$) with two different test batch sizes, 500 and 1000. Table 8 demonstrates that the upper-bounds for both batch

**Figure 2:** Left: the sigmoid function. Right: choice of hyper-parameter gamma in soft indicator function for set $I = [0.45, 0.55]$.

**Table 7:** Exact D-Cal, Soft-Dcal, and NLL at end of training, evaluated on training data for models trained with $\lambda = 10$ and batch size $1,000$. Approximation improves as $\gamma$ increases. Gradients vanish when $\gamma$ gets too large. All experiments are better in calibration than the $\lambda = 0$ MLE model, which has exact D-cal 0.09.

| $\gamma$ | 10 | $10^2$ | $10^3$ | $10^4$ | $10^5$ | $10^6$ | $10^7$ | $5 \times 10^7$ |
|---|---|---|---|---|---|---|---|---|
| Exact D-Cal | 0.2337 | 0.0095 | 0.0079 | 0.0039 | 0.0025 | 0.0014 | 0.0015 | 0.0048 |
| Soft D-Cal | 0.4599 | 0.0604 | 0.0074 | 0.0039 | 0.0025 | 0.0014 | 0.0015 | 0.0048 |
| NLL | 2.1180 | 1.1362 | 1.0793 | 1.2508 | 1.6993 | 2.3873 | 2.6940 | 3.4377 |

sizes preserve model ordering with respect to exact D-CALIBRATION. The bound for batch size $10,000$ is quite close to the exact D-CALIBRATION.

# I Modification of soft indicator for the first and the last interval

In our soft indicator,

$$\zeta_\gamma(u; I) = \text{Sigmoid}(\gamma(u - a)(b - u)) = (1 + \exp(-\gamma(u - a)(b - u)))^{-1}$$

is a differentiable approximation for $\mathbb{1}\left[u \in [a, b]\right]$. When $b$ is the upper boundary of all the $u$ values, for example, 1 for CDF values, the $b$ in the soft indicator can be replaced by any value that is greater than $b$. We use 2 to replace 1 for the upper boundary when $b = 1$ in our experiments. Similarly we use $a = -1$ to replace $a = 0$ for the lower boundary when $a = 0$.

**Table 8:** Slack in the upper-bound preserves modeling ordering with respect to exact D-CALIBRATION

| $\lambda$ | Batch Size | Exact D-Cal | Upper-bound |
|---|---|---|---|
| 0 | 500 | 0.05883 | 0.0605 |
|   | 10000 | " | 0.0589 |
| 1 | 500 | 0.02204 | 0.0238 |
|   | 10000 | " | 0.0221 |
| 5 | 500 | 0.00963 | 0.0114 |
|   | 10000 | " | 0.0097 |
| 10 | 500 | 0.00482 | 0.0066 |
|   | 10000 | " | 0.0048 |
| 50 | 500 | 0.00040 | 0.0021 |
|   | 10000 | " | 0.0004 |
| 100 | 500 | 0.00022 | 0.0021 |
|   | 10000 | " | 0.0003 |
| 500 | 500 | 0.00015 | 0.0020 |
|   | 10000 | " | 0.0002 |
| 1000 | 500 | 0.00006 | 0.0019 |
|   | 10000 | " | 0.0001 |

Consider the term in our upper-bound (eq. (6)) for the last interval $I = [a, b]$, where $b = 1$, $\left(\frac{1}{M} \sum_i \zeta_\gamma(u_i; I) - |I|\right)^2$. The gradient of this term with respect to one CDF value $u_i$ is:

$$\frac{d}{du_i}\left(\frac{1}{M}\sum_i \zeta_\gamma(u_i; I) - |I|\right)^2$$

$$= \frac{d}{du_i}\left(\frac{1}{M}\sum_i \text{Sigmoid}(\gamma(u_i - a)(b - u_i)) - |I|\right)^2$$

$$\left[\text{let } A := 2/M * \left(\frac{1}{M}\sum_i \text{Sigmoid}(\gamma(u_i - a)(b - u_i)) - |I|\right)\right]$$

$$= A\frac{d}{du_i}\text{Sigmoid}(\gamma(u_i - a)(b - u_i))$$

$$= A * -\frac{\exp(-\gamma(u_i - a)(b - u_i))}{(1 + \exp(-\gamma(u_i - a)(b - u_i)))^2}\frac{d}{du_i}(-\gamma(u_i - a)(b - u_i))$$

$$= A * \frac{\exp(-\gamma(u_i - a)(b - u_i))}{(1 + \exp(-\gamma(u_i - a)(b - u_i)))^2} * \gamma * (a + b - 2u_i)$$

If

$$\frac{1}{M}\sum_i \zeta_\gamma(u_i; I) - |I| > 0,$$

then the fraction of points in the interval is larger than the size of the interval. We want to move the points out of the interval. In the last interval, in order to move points out of the interval, we can only make the values smaller, which means we want the gradient with respect to $u$ to be positive. (recall that we are moving in the direction of the negative gradient to minimize the objective). However, for points that are greater than $(a + b)/2$, the above gradient will be negative because term $(a + b - 2u_i)$ is negative. This is not ideal. Changing the value $b$ from 1 to 2 can resolve the issue. Since CDF values are all smaller than 1, $(a + b)/2$ will always be greater than $u$ if we use $b = 2$ for the last interval. The above optimization issue only applies on the first and last interval because for intervals in the middle, we can move the points either to left or right to lower the fraction of points in the interval.

# J  Full Results: More Models and Choices of Lambda

**Table 9:** Gamma simulation, uncensored (full results)

|        |        | $\lambda$ | 0 | 1 | 5 | 10 | 50 | 100 | 500 | 1000 |
|--------|--------|---|---|---|---|----|----|-----|-----|------|
| Log-Norm | NLL | | 0.381 | 0.423 | 0.507 | 0.580 | 0.763 | 0.809 | 0.870 | 0.882 |
| NLL | D-CAL | | 0.271 | 0.060 | 0.021 | 0.011 | 0.001 | 4e-4 | 1e-4 | 7e-5 |
|  | CONC | | 0.982 | 0.955 | 0.931 | 0.908 | 0.841 | 0.835 | 0.809 | 0.802 |
| Log-Norm | NLL | | 0.455 | 0.614 | 0.730 | 0.781 | 0.837 | 0.848 | 0.869 | 0.965 |
| S-CRPS | D-CAL | | 0.055 | 0.014 | 0.004 | 0.002 | 2e-4 | 1e-4 | 1e-4 | 1e-4 |
|  | CONC | | 0.979 | 0.975 | 0.968 | 0.959 | 0.940 | 0.931 | 0.864 | 0.811 |
| Cat-NI | NLL | | 0.998 | 1.042 | 1.129 | 1.197 | 1.788 | 2.098 | 3.148 | 3.688 |
|  | D-CAL | | 0.074 | 0.023 | 0.008 | 0.005 | 4e-4 | 4e-4 | 2e-4 | 1e-4 |
|  | CONC | | 0.986 | 0.986 | 0.985 | 0.985 | 0.973 | 0.960 | 0.877 | 0.748 |
| Cat-I | NLL | | 0.997 | 1.001 | 1.029 | 1.083 | 1.763 | 2.083 | 3.167 | 3.788 |
|  | D-CAL | | 0.002 | 0.002 | 0.001 | 0.002 | 5e-4 | 5e-4 | 1e-4 | 1e-4 |
|  | CONC | | 0.986 | 0.986 | 0.986 | 0.985 | 0.972 | 0.960 | 0.874 | 0.699 |
| MTLR-NI | NLL | | 1.287 | 1.409 | 1.589 | 1.612 | 2.356 | 2.590 | 3.267 | 3.509 |
|  | D-CAL | | 0.027 | 0.027 | 0.015 | 0.008 | 5e-4 | 2e-4 | 2e-4 | 2e-4 |
|  | CONC | | 0.986 | 0.986 | 0.983 | 0.981 | 0.952 | 0.940 | 0.909 | 0.899 |
| MTLR-I | NLL | | 1.392 | 1.419 | 1.616 | 1.823 | 2.165 | 2.612 | 2.982 | 3.184 |
|  | D-CAL | | 0.048 | 0.034 | 0.017 | 0.009 | 7e-4 | 2e-4 | 1e-4 | 1e-4 |
|  | CONC | | 0.986 | 0.986 | 0.982 | 0.980 | 0.958 | 0.934 | 0.918 | 0.917 |

**Table 10:** Gamma simulation, censored (full results). For categorical model with interpolation, the D-CAL is already very low at $\lambda = 0$ so it is hard to optimize this one further.

|        |        | $\lambda$ | 0 | 1 | 5 | 10 | 50 | 100 | 500 | 1000 |
|--------|--------|---|---|---|---|----|----|-----|-----|------|
| Log-Norm | NLL | | -0.059 | -0.049 | -0.022 | 0.004 | 0.099 | 0.138 | 0.191 | 0.215 |
| NLL | D-CAL | | 0.029 | 0.020 | 0.008 | 0.005 | 7e-4 | 2e-4 | 6e-5 | 7e-5 |
|  | CONC | | 0.981 | 0.969 | 0.950 | 0.942 | 0.927 | 0.916 | 0.914 | 0.897 |
| Log-Norm | NLL | | 0.038 | 0.084 | 0.119 | 0.143 | 0.185 | 0.201 | 0.343 | 0.436 |
| S-CRPS | D-CAL | | 0.017 | 0.007 | 0.003 | 0.001 | 1e-4 | 1e-4 | 5e-5 | 8e-5 |
|  | CONC | | 0.982 | 0.978 | 0.971 | 0.963 | 0.952 | 0.950 | 0.850 | 0.855 |
| Cat-NI | NLL | | 0.797 | 0.799 | 0.805 | 0.822 | 1.023 | 1.149 | 1.665 | 1.920 |
|  | D-CAL | | 0.009 | 0.006 | 0.003 | 0.002 | 3e-4 | 2e-4 | 6e-5 | 6e-5 |
|  | CONC | | 0.987 | 0.987 | 0.987 | 0.987 | 0.982 | 0.976 | 0.922 | 0.861 |
| Cat-I | NLL | | 0.783 | 0.782 | 0.788 | 0.795 | 0.948 | 1.124 | 1.686 | 1.994 |
|  | D-CAL | | 7e-5 | 1e-4 | 6e-5 | 8e-5 | 2e-4 | 2e-4 | 4e-5 | 6e-5 |
|  | CONC | | 0.987 | 0.987 | 0.987 | 0.987 | 0.983 | 0.976 | 0.933 | 0.847 |
| MTLR-NI | NLL | | 0.873 | 0.875 | 0.875 | 0.977 | 1.271 | 1.412 | 1.747 | 1.900 |
|  | D-CAL | | 0.004 | 0.004 | 0.003 | 0.004 | 4e-4 | 2e-4 | 2e-4 | 2e-4 |
|  | CONC | | 0.987 | 0.987 | 0.987 | 0.985 | 0.973 | 0.965 | 0.951 | 0.943 |
| MTLR-I | NLL | | 0.829 | 0.830 | 0.866 | 0.981 | 1.266 | 1.414 | 1.762 | 1.912 |
|  | D-CAL | | 0.004 | 0.004 | 0.004 | 0.004 | 5e-4 | 1e-4 | 6e-5 | 7e-5 |
|  | CONC | | 0.988 | 0.988 | 0.987 | 0.985 | 0.971 | 0.963 | 0.947 | 0.939 |

**Table 11:** Survival-MNIST, uncensored (full results)

|  |  λ | 0 | 1 | 5 | 10 | 50 | 100 | 500 | 1000 |
|---|---|---|---|---|---|---|---|---|---|
| Log-Norm NLL | NLL | 4.344 | 4.407 | 4.530 | 4.508 | 4.549 | 4.571 | 5.265 | 5.417 |
|  | D-CAL | 0.328 | 0.104 | 0.018 | 0.020 | 0.011 | 0.010 | 0.005 | 0.005 |
|  | CONC | 0.886 | 0.867 | 0.754 | 0.759 | 0.725 | 0.713 | 0.541 | 0.509 |
| Log-Norm S-CRPS | NLL | 4.983 | 4.940 | 4.853 | 4.759 | 4.714 | 4.673 | 4.852 | 5.118 |
|  | D-CAL | 0.212 | 0.132 | 0.081 | 0.059 | 0.020 | 0.007 | 0.003 | 0.003 |
|  | CONC | 0.889 | 0.878 | 0.866 | 0.861 | 0.873 | 0.873 | 0.820 | 0.798 |
| Cat-NI | NLL | 1.726 | 1.730 | 1.737 | 1.755 | 1.824 | 1.860 | 2.076 | 3.073 |
|  | D-CAL | 0.019 | 0.013 | 0.008 | 0.005 | 9e-4 | 9e-4 | 6e-4 | 3e-4 |
|  | CONC | 0.945 | 0.945 | 0.945 | 0.937 | 0.921 | 0.916 | 0.854 | 0.690 |
| Cat-I | NLL | 1.726 | 1.731 | 1.735 | 1.741 | 1.782 | 1.809 | 1.953 | 2.157 |
|  | D-CAL | 0.007 | 0.005 | 0.003 | 0.002 | 6e-4 | 3e-4 | 4e-4 | 3e-4 |
|  | CONC | 0.945 | 0.945 | 0.945 | 0.945 | 0.940 | 0.937 | 0.897 | 0.830 |
| MTLR-NI | NLL | 1.747 | 1.745 | 1.749 | 1.772 | 1.832 | 1.850 | 2.075 | 2.419 |
|  | D-CAL | 0.018 | 0.014 | 0.008 | 0.004 | 0.001 | 0.001 | 8e-4 | 0.002 |
|  | CONC | 0.944 | 0.945 | 0.945 | 0.944 | 0.934 | 0.934 | 0.870 | 0.808 |
| MTLR-I | NLL | 1.746 | 1.746 | 1.752 | 1.756 | 1.779 | 1.802 | 1.975 | 2.560 |
|  | D-CAL | 0.005 | 0.004 | 0.003 | 0.002 | 5e-4 | 4e-4 | 8e-4 | 0.001 |
|  | CONC | 0.944 | 0.944 | 0.945 | 0.944 | 0.941 | 0.936 | 0.886 | 0.806 |

**Table 12:** Survival-MNIST, censored (full results)

|  |  λ | 0 | 1 | 5 | 10 | 50 | 100 | 500 | 1000 |
|---|---|---|---|---|---|---|---|---|---|
| Log-Norm NLL | NLL | 4.337 | 4.377 | 4.433 | 4.483 | 4.602 | 4.682 | 4.914 | 5.151 |
|  | D-CAL | 0.392 | 0.074 | 0.033 | 0.020 | 0.008 | 0.005 | 0.005 | 0.007 |
|  | CONC | 0.902 | 0.873 | 0.829 | 0.794 | 0.728 | 0.696 | 0.628 | 0.573 |
| Log-Norm S-CRPS | NLL | 4.950 | 4.929 | 4.873 | 4.859 | 4.672 | 4.749 | 4.786 | 4.877 |
|  | D-CAL | 0.215 | 0.122 | 0.071 | 0.051 | 0.018 | 0.010 | 0.002 | 9e-4 |
|  | CONC | 0.891 | 0.881 | 0.871 | 0.874 | 0.866 | 0.868 | 0.839 | 0.815 |
| Cat-NI | NLL | 1.733 | 1.734 | 1.738 | 1.765 | 1.827 | 1.861 | 2.074 | 3.030 |
|  | D-CAL | 0.018 | 0.014 | 0.008 | 0.004 | 8e-4 | 5e-4 | 5e-4 | 4e-4 |
|  | CONC | 0.945 | 0.945 | 0.944 | 0.927 | 0.920 | 0.919 | 0.862 | 0.713 |
| Cat-I | NLL | 1.731 | 1.731 | 1.741 | 1.750 | 1.779 | 1.805 | 1.955 | 2.113 |
|  | D-CAL | 0.007 | 0.006 | 0.003 | 0.002 | 3e-4 | 4e-4 | 4e-4 | 3e-4 |
|  | CONC | 0.945 | 0.944 | 0.945 | 0.945 | 0.942 | 0.938 | 0.901 | 0.843 |
| MTLR-NI | NLL | 1.126 | 1.118 | 1.125 | 1.136 | 1.174 | 1.193 | 1.350 | 1.482 |
|  | D-CAL | 0.021 | 0.017 | 0.012 | 0.009 | 0.006 | 0.006 | 0.006 | 0.007 |
|  | CONC | 0.958 | 0.960 | 0.961 | 0.960 | 0.949 | 0.943 | 0.897 | 0.880 |
| MTLR-I | NLL | 1.126 | 1.118 | 1.125 | 1.136 | 1.174 | 1.193 | 1.350 | 1.482 |
|  | D-CAL | 0.021 | 0.017 | 0.012 | 0.009 | 0.006 | 0.006 | 0.006 | 0.007 |
|  | CONC | 0.958 | 0.960 | 0.961 | 0.960 | 0.949 | 0.943 | 0.897 | 0.880 |

**Table 13:** MIMIC-III length of stay (full results)

|  | $\lambda$ | 0 | 1 | 5 | 10 | 50 | 100 | 500 | 1000 |
|---|---|---|---|---|---|---|---|---|---|
| Log-Norm | D-CAL | 0.860 | 0.639 | 0.210 | 0.155 | 0.066 | 0.046 | 0.009 | 0.005 |
| S-CRPS | CONC | 0.625 | 0.639 | 0.577 | 0.575 | 0.558 | 0.555 | 0.528 | 0.506 |
| Cat-NI | NLL | 3.142 | 3.177 | 3.101 | 3.167 | 3.086 | 3.088 | 3.448 | 3.665 |
|  | D-CAL | 0.002 | 0.002 | 0.002 | 0.001 | 3e-4 | 2e-4 | 1e-4 | 1e-4 |
|  | CONC | 0.702 | 0.700 | 0.701 | 0.699 | 0.695 | 0.690 | 0.642 | 0.627 |
| Cat-I | NLL | 3.142 | 3.075 | 3.157 | 3.073 | 3.002 | 3.073 | 3.364 | 3.708 |
|  | D-CAL | 4-e4 | 3e-4 | 3e-4 | 3e-4 | 4e-4 | 1e-4 | 5e-5 | 4e-5 |
|  | CONC | 0.702 | 0.702 | 0.701 | 0.702 | 0.698 | 0.695 | 0.638 | 0.627 |

**Table 14:** The Cancer Genome Atlas, glioma (full results)

|  | $\lambda$ | 0 | 1 | 5 | 10 | 50 | 100 | 500 | 1000 |
|---|---|---|---|---|---|---|---|---|---|
| Weibull | NLL | 4.436 | 4.390 | 4.313 | 4.292 | 4.441 | 4.498 | 4.475 | 4.528 |
|  | D-CAL | 0.035 | 0.028 | 0.014 | 0.009 | 0.003 | 0.003 | 0.004 | 0.007 |
|  | CONC | 0.788 | 0.785 | 0.781 | 0.777 | 0.731 | 0.702 | 0.608 | 0.575 |
| Log-Norm | NLL | 14.187 | 6.585 | 4.841 | 4.639 | 4.181 | 4.181 | 4.403 | 4.510 |
| NLL | D-CAL | 0.059 | 0.024 | 0.012 | 0.010 | 0.003 | 0.003 | 0.002 | 0.004 |
|  | CONC | 0.657 | 0.632 | 0.673 | 0.703 | 0.778 | 0.805 | 0.474 | 0.387 |
| Log-Norm | NLL | 5.784 | 5.801 | 5.731 | 5.698 | 5.047 | 4.892 | 4.750 | 4.712 |
| S-CRPS | D-CAL | 0.258 | 0.2585 | 0.257 | 0.252 | 0.100 | 0.0702 | 0.044 | 0.025 |
|  | CONC | 0.798 | 0.798 | 0.798 | 0.810 | 0.568 | 0.507 | 0.420 | 0.363 |
| Cat-NI | NLL | 1.718 | 1.742 | 1.746 | 1.758 | 1.800 | 1.799 | 1.810 | 1.826 |
|  | D-CAL | 0.008 | 0.003 | 0.002 | 0.002 | 0.003 | 0.003 | 0.003 | 0.002 |
|  | CONC | 0.781 | 0.771 | 0.775 | 0.775 | 0.765 | 0.765 | 0.758 | 0.748 |
| Cat-I | NLL | 1.711 | 1.718 | 1.733 | 1.726 | 1.743 | 1.787 | 1.781 | 1.789 |
|  | D-CAL | 0.003 | 0.001 | 8e-4 | 0.001 | 0.002 | 0.002 | 0.002 | 0.002 |
|  | CONC | 0.778 | 0.779 | 0.780 | 0.798 | 0.804 | 0.803 | 0.806 | 0.802 |
| MTLR-NI | NLL | 1.624 | 1.620 | 1.636 | 1.636 | 1.666 | 1.658 | 1.748 | 1.758 |
|  | D-CAL | 0.009 | 0.007 | 0.007 | 0.005 | 0.003 | 0.003 | 0.002 | 0.002 |
|  | CONC | 0.828 | 0.829 | 0.822 | 0.824 | 0.814 | 0.818 | 0.788 | 0.763 |
| MTLR-I | NLL | 1.616 | 1.626 | 1.612 | 1.612 | 1.632 | 1.640 | 1.636 | 1.753 |
|  | D-CAL | 0.003 | 0.003 | 0.002 | 0.001 | 0.001 | 0.001 | 9e-4 | 0.001 |
|  | CONC | 0.827 | 0.825 | 0.831 | 0.829 | 0.824 | 0.823 | 0.825 | 0.783 |

## Footnotes

[5]https://www.cancer.gov/about-nci/organization/ccg/research/structural-genomics/tcga/studied-cancers/glioma