[Reviews · NeurIPS 2020]

Review 1

Summary and Contributions: The authors provide a novel extension to an evaluation metric in the area of survival production known as distribution calibration (D-Calibration). Their extension, X-Calibration (X-Cal) is differentiable and can be used as a regularizer in the objective function to control the degree of calibration when training a model. The authors then compare a neural network trained using this regularizer to a S-CRPS model and show how the calibration can be tuned with one additional hyper-parameter, achieving much better calibration scores (measured with D-Calibration). This is an important contribution tothe field of survival prediction: while many metrics/improvements solely focus on discrimination and optimizing the conconcordance index, this method allows for that type of optimization but also adds a straightforward regularization approach to incorporate a calibration measure.

Strengths: This work is very strong, as it has a mathematically sound extension of D-Calibration that takes the previous work from being a post-hoc test for calibration to actively addressing calibration in the training stage via regularization. As previously mentioned, much of the field focuses on maximizing discriminatory metrics -- e.g. concordance -- but this work is an important addition to the field as it addresses the calibration angle of survival prediction. The authors also provide an extensive empirical analysis that supports their results, showing that one can tune their model’s level of calibration with X-Cal. The authors only compare against one other model, which may be seen as a weakness, but their contribution comes across strongly that they can increase/decrease the calibration level with their hyperparameter selection.

Weaknesses: There are no glaring weaknesses in this paper. Smaller issues -- it would be useful to test their model on a real-world dataset that includes censoring, since this is the primary challenge of survival prediction. The related work section feels choppy but in terms of content it appears to be valid and well placed -- nothing appears to be left out. Rebuttal: Minor concern that the authors did not address the other reviewer's concerns that other metrics are not included (1-Calibration). But this not major, as optimizing for a "distributional (D)-Calibration" was the goal of the manuscript and is shown throughout the paper and since (as the author mentions) it is known that optimizing D-Calibration does not necessarily optimize for 1-Calibration. So it doesn't feel like this is an explicit comparison needed to be made. [This Weakness box previously mistakenly mischaracterized the authors rebuttal's comparison to MTLR. This was incorrect, and has been removed. This review's score is back to the original "8".]

Correctness: There were no apparent issues with claims or theoretical/empirical methodology.

Clarity: I had no issues following this paper, I find it very easy to follow and feel it is well organized. My brief issue was that the ending felt rushed, the related work section felt choppy and the Broader Impact section could be expanded.

Relation to Prior Work: The connections with prior work (primarily D-Calibration) were well discussed and is self-contained enough that there are no misunderstandings. The extension from D-Calibration to X-Calibration is made apparent and so differs from any previous work.

Reproducibility: Yes

Additional Feedback: Line 102 - Should explicitly state that |C| = b-a; it took me a minute to realize this. Line 133 - It may be helpful to walk through this estimation in the Appendix; I found the estimation of (6) non-obvious. Line 177 - I would recommend writing \gamma = 10000, rather than 10k, as this suggests that k is a parameter. It would be helpful if the code (for the empirical analysis) were available in a public github repository or some other format. Googling the title of the main reference [Haider 2018], I found the JMLR http://www.jmlr.org/papers/v21/18-772.html, which appears to be the same material. Probably better to use this updated citation.


Review 2

Summary and Contributions: The authors present X-CAL, an approach that turns D-Calibration into a differentiable regularizer that can be added to existing survival analysis objective functions thus allowing for trading-off predictive power and calibration. Experiments on real and artificial data suggest that their proposed approach delivers improved D-Calibration without a large log-likelihood penalty.

Strengths: The authors propose a simple continuous relaxation of the D-Calibration measure of Haider et al. (2018) amenable to maximum likelihood estimation.

Weaknesses: It is not clear why the authors do not show results for other survival analysis approaches that though not explicitly addressing calibration often result in calibrated results according to Haider et al. (2018). Moreover, MTLR which is closely related to the proposed approach, in that they bin time, is not even referenced. The authors only present results in a real dataset when there are plenty of publicly available and well-known datasets for survival analysis benchmarks. It is not clear why the authors choose to present D-Calibration results as opposed to the (p-value of the) test statistic in Haider et al. (2018). The comments above being considered, the experiments section of the paper is largely underwhelming.

Correctness: The formulation for the proposed approach, largely supported by results in Haider et al. (2018) seems correct.

Clarity: The paper is well-written, the background is sufficient, the proposed approach is clearly introduced and the description of the experimental setup is sufficient and complemented well in the supplementary material.

Relation to Prior Work: The authors concisely review the related work and present calibration and D-calibration, more specifically, as background for their work.

Reproducibility: Yes

Additional Feedback: k in line 177 is not defined. The authors mention that the slack in their bound does not affect the experiments which is reasonable, however, it would be interesting to see the bound values in relation to the exact D-Calibration results. It would be interesting to see results for different values of \gamma. It would be good if the authors provide some guidance into how to select \lambda. The authors have addressed my questions about the p-value, the bound, \gamma and \lambda in their rebuttal.


Review 3

Summary and Contributions: This paper considers a new way to regularize machine learning methods with survival (ie right-censored, non-negative) outcomes to be better calibrated.

Strengths: Calibration is a really important problem in training neural networks and in survival analysis problems. This paper goes beyond evaluating or post-hoc adjusting calibration to regularizing towards calibrated models during training.

Weaknesses: It's not clear exactly what happens when the learned conditional survival function is a discontinuous, as is often the case in models based on the conditional Kaplan-Meier estimator or other nonparametric estimates of the hazard function based on the observed data. Given the prevalence of these in survival analysis, a discussion of how this notion of calibration can (or cannot) be used is important. This notion of calibration is hard to connect to existing notions, and the experiments do not address this. While the notion of calibration explored here is over all time points, it would be helpful to compare models in terms of more established notions of calibration at a couple of fixed time points, such as the one in suggested for survival outcomes in Yadlowsky, et al. "A calibration metric for risk scores with survival data." Machine Learning for Healthcare Conference. 2019. Edit: While it makes sense that calibration over many times doesn't necessarily imply calibration at a specific time (1-calibration, as referred), it seems like if one were to check calibration at a few time points, that would help clarify when the model is well-calibrated versus not. The lack of connection to well understood notions of calibration is a big weakness of this paper. I would question use of this regularizer if it led to poorer 1-calibration at a number of equally spaced, or clinically significant, times.

Correctness: Very few concrete claims are made in the paper. As far as I can tell, all mathematical statements are true with enough conditions (such as continuity and various other assumptions), but many things are given heuristically. In particular, they make some adjustments for right censoring, however it's unclear if these recover the correct calibration estimate, consistent with what the calibration would've been if the censored outcomes were revealed. I'm concerned this is not true, given that the reported calibration is much better in the censored MNIST experiment than the uncensored one. Given that the motivation of the paper is to address calibration in survival analysis, some concrete justification of the suitability of the censoring adjustment presented in the paper is necessary.

Clarity: The paper is well-written, however would benefit for more precise and less heuristic statements about mathematical results and connections to prior literature. Edit: To emphasize the point made below re: prior work, this paper does not seem to stand alone, but depends heavily on Haider et al., 2018. For example, many of the responses to reviewer questions were directed at that paper. While there is no need to reinvent the wheel, the paper would be much improved if it did a better job articulating the concepts than just pointing to another paper.

Relation to Prior Work: It was really hard to tell where the results of Haider et al. [2018] stop and the results of this paper begin. The two are so intertwined it is not very clear what are actual contributions of this paper. The paper would benefit by evaluating calibration using some other notions of calibration explored in the survival analysis literature, as noted above.

Reproducibility: Yes

Additional Feedback: Responses to author feedback: The authors did not adequately address many of my concerns, or those of other reviewers asking for more careful comparison to other approaches. Given that trivially calibrated models exist (for example, the population Kaplan-Meier estimator that doesn't use any covariate information), arguing that one improved calibration without hurting discrimination requires careful evaluation and comparison to other approaches. The paper would be much improved, and I encourage the authors to add the suggested evaluations by all reviewers, as they seem to be reasonable requests. Without these evaluations, I do not feel that the contribution of this method is articulated well enough, and am changing my overall score to a 4.


Review 4

Summary and Contributions: The authors propose a new regularization term, D-calibration, that can be added to the loss when training a discrete-time survival model, and enforces good calibration of the predicted survival times on the training set. Instead of assessing calibration by comparing predicted vs. actual survival proportion at a specific follow-up time point, this measure looks at whether the correct proportion of individuals' failure times fall into the various intervals of the predicted cumulative distribution function of time. The authors find that by varying the regularization strength, they can achieve varying tradeoffs between discrimination (measured by C-index) and calibration, and that their method outperforms a competing method, Survival-CRPS on clinical (MIMIC) and synthetic datasets including survival MNIST.

Strengths: The claims seem generally sound. The method is novel and clever. I can see myself using it in my own applied work. A big advantage of the approach is that the calibration score evaluates calibration through the entire follow-up time, not just at one follow-up time point.

Weaknesses: Any kind of predictive model, and especially deep neural networks, will tend to overfit to the training set, generally causing predictions on a separate test set to be too extreme (shrinkage, or calibration slope of less than 1). The authors' X-cal procedure ensures good calibration on the training set. But that could result in disappointing calibration when applied to the test set. It seems to me that one would want a procedure to maximize calibration on a validation set, not the training set. That would then lead to good calibration on the separate test set. An example is picking regularization strength (L2 regularization, dropout, etc.). One picks the strength that maximizes likelihood or calibration on the validation set, not training set. The authors do not seem to address this issue directly (though they allude to it a bit in the Post-Training Calibration Methods section, line 265). In the experiments, calibration performance was assessed only with D-calibration. This is fine, but since the neural network was optimized with X-cal, something very similar to D-calibration, it is not surprising that use of X-cal had better D-calibration than the competing S-CRPS method. I would also find it useful to see some other calibration assessments comparing D-calibration to S-CRPS. For instance, one can make a calibration plot as per Royston & Altman, External validation of a Cox prognostic model: principles and methods. First, group the test set patients into 10 bins of predicted probability of >7 day ICU stay (I am not too familiar with MIMIC so perhaps a different number would be more appropriate). For each bin, calculated the predicted survival proportion at 7 days, and overlay the corresponding value from the actual Kaplan-Meier survival curve. A well-calibrated model will have similar predicted and actual values for each bin. [update after authors' responses: The authors refer to this sort of calibration as 1-calibration and state that their focus is not to optimize this metric. I understand that this is not their focus, but showing the 1-calibration results would be useful to put the work into context of past studies and methods, and 1-calibration is much more established than D-calibration. Showing that the authors' method results in decent 1-calibration at a few relevant time points would be a sort of sanity check.] It does not appear that implementation code was released. As the implementation seems somewhat complex in terms of dealing with censored data, this would make it harder to use the method in one's own work. [update after authors' responses: the authors now state the code will be released on GitHub which is good.] [New comment from reading the paper another time]: Table 5 shows that on the real-world MIMIC dataset, something went very wrong with the comparison method S-CRPS (C-index of 0.499, i.e. no better than null model). So, there is basically no baseline/comparison result to benchmark X-cal against for the real-world data. Please consider adding results for another publicly available neural network survival model such as DeepSurv (I'm not as familiar with MTLR but if that could be applied to MIMIC then that would be fine too).

Correctness: The methods seem to be correct.

Clarity: Yes.

Relation to Prior Work: Yes.

Reproducibility: Yes

Additional Feedback: Line 38: "Calibration means accurate prognosis, which prevents giving patients misinformed limits on their survival, such as 6 months when they would survive years." Consider rephrasing. The Kaplan-Meier estimator with no predictor variables is well calibrated, but would not perform well when making survival predictions for individual patients since it would ignore predictor data. Concordance: When the authors refer to concordance, I assume they are referring to Harrell's concordance index, but it would be helpful to define concordance on first use.

[Author Response · NeurIPS 2020]

We thank the reviewers for their feedback. We will release our implementation on github. We thank R1 for pointing out
our calibration metric of choice, D-Calibration (D-Cal), has been published at JMLR 2020 (we cite this as [H] here).

**[R3: Relation to D-Cal]** Building on [H]'s D-cal, we propose X-cal to regularize a model to have low D-Cal. X-cal is
a differentiable approximation of an upper-bound on D-Cal, amenable to stochastic optimization.

**[R1, R2: Real world dataset with censoring; Survival benchmarks]** We evaluated X-Cal on [Avati et al]'s alternative
MIMIC set with $70\%$ censored points. D-Cal goes from $2 \times 10^{-4} \to 9 \times 10^{-5}$ as $\lambda$ increases from $0 \to 10^3$. We will
add this to the paper. We are glad to include specific evaluations/benchmark sets that the reviewers think are relevant.

**[R2: Comparison with MTLR/ approaches mentioned in [H]]** We do not include methods from [H] because our
work focuses on using flexible models with good likelihood but poor calibration, like S-CRPS. We use a categorical
model because it is a common flexible likelihood that can approximate many continuous distributions given enough bins.
This allows us to evaluate X-Cal without parametric restrictions. We did cite an approach like MTLR [Ranganath 2018].
We will cite MTLR [Yu et al. 2011] and its neural version [Fotso 2018]. We ran MTLR using PyCox library on the
uncensored synthetic gamma dataset. This gave a model with D-cal $0.7486$, which is higher than any model we study.

**[R3,R4: Comparison to other established calibration metrics]** The alternative notion of calibration for fixed time t*
suggested by reviewers [Yadlowsky 2019, Royston/Altman 2013] are described in [H] as "1-Calibration". [H] proves
that D-Cal (with fixed bins) and 1-Cal for time t* (with fixed bins) measure different aspects of the survival distribution:
0 D-Cal and 0 1-Cal do not imply each other. A practitioner may need calibration at several times e.g. 6 months, 1 year.
Future work is to regularize models with approximations of 1-Cal. measures (e.g. Hosmer-Lemeshow statistic) using
soft indicators. Our focus is to maintain a certain level of calibration based on the specific metric, D-cal.

**[R2: $p$-value]** The $p$-value reported by [H] is the result of a $\chi^2$-test on the D-Cal test statistic. Thus, if models are
ordered in the test statistic their corresponding $p$-values are ordered in the same way. While $p$-values help test for perfect
calibration, our focus is on *improving* calibration of existing models which we demonstrate in in our experiments.

**[R3: Discontinuous learned conditional survival model]** As mentioned in 4.1, a discontinuous model will have a
lower bound greater than 0 for D-Cal because its CDF values will not be a continuous Unif(0,1) variable. However,
minimizing D-Cal will still spread out the CDF values to whichever extent possible and thus improve calibration. In the
case of a categorical model, this lower bound decreases to 0 as the size of each bin goes to $0$ when adding more bins.

**[R3: Adjustments for right censoring / MNIST censoring]** This is an important issue. In line 151 of our paper, we
handle right censoring using the technique proposed in appendix B.5 in [H] and proved to result in a valid test statistic.
As noted in [H] on page 47, the estimation of D-cal on a censored dataset will not equal the estimate when censored
times are revealed. This is due to the fact that in the censored dataset [H]'s correction for right censoring gives a
few bins the correct weight for free meaning D-cal will be lesser. However, for a given dataset, D-Cal is 0 for any
bin for the true conditional $p(T \mid X)$ for any non-informative censoring process that meets a "positivity" assumption.
Thus, two models evaluated on the *same* data (censored or uncensored) can be compared with D-Cal. Reweighting
methods, such as Yadlowsky et al. that R3 suggests, can be used to adjust for censoring. One option is to adjust with
with $p(C \mid X)$. This requires $C \perp T \mid X$ and solving a censored survival problem $p(C \mid X)$ with a high-dimensional
conditioning set. Another option is to adjust with the lower dimensional conditioning set $p(C \mid risk_\theta(X))$. This
requires $C \perp T \mid risk_\theta(X)$ and differentiating through the *estimation* of $p(C \mid risk_\theta(X))$ w.r.t. $\theta$. The approach we
take requires neither another censored survival problem nor differentiating through estimation.

**[R2: $\lambda$ and $\gamma$]** Choosing $\lambda$: the user first decides on a
threshold of D-Cal and then increase $\lambda$ until D-Cal eval-
uated on a held-out validation set meets this threshold.
See Table 1 for the role of $\gamma$. With low $\gamma$, soft d-cal approximates
poorly and D-Cal/NLL suffer. For $\gamma$ too large, gradients vanish
and D-Cal/NLL suffer. We found $\gamma = 10^4$ allowed for easy
optimization with soft D-Cal approximating D-Cal well.

| $\gamma$ | 10 | $10^4$ | $1.1 \times 10^4$ | $10^5$ |
|---|---|---|---|---|
| D-Cal | 0.4 | 0.0005 | 0.0002 | 0.0003 |
| NLL | 4.33 | 1.82 | 1.88 | 2.49 |

Table 1: Soft D-cal as $\gamma$ varies.

**[R2, Tightness of upper-bound]** Table 2 shows that models ordered by
the upper-bound are ordered in D-cal the same way. Further, when batch
size is large enough, if $\lambda_i < \lambda_j$, the bound for $\lambda_j$ is less than D-Cal for $\lambda_i$.

**[R4: Choosing D-Cal on a validation set]** During training we evaluated
NLL+D-cal on a validation set at every epoch and save the model. Then,
we report test metrics for the model with best validation NLL + D-Cal. If
we only select/optimize X-cal on a validation set, the predictive likelihood
may get arbitrarily worse. This issue occurs with Platt scaling as well.

| $\lambda$ | Batch size | D-Cal | Bound |
|---|---|---|---|
| 10 | 500 | 0.0040 | 0.0059 |
|  | 5000 | " | 0.0042 |
| 50 | 500 | 0.0006 | 0.0024 |
|  | 5000 | " | 0.0008 |
| 100 | 500 | 0.0003 | 0.0022 |
|  | 5000 | " | 0.0005 |

Table 2: Slack in the Upper Bound

**[Minor comments/ Typos]** We thank the reviewers for detailed feedback
about our writing. We will define Harrell's Concordance Index, change 10k
to our intended $10^4$, and rephrase "calibration means accurate prognosis".

[Meta-Review · NeurIPS 2020]

For survival analysis, where calibrated models obviously are important, this paper introduces a differentiable plug-and-play regularizer which allows optimizing calibration, and choosing a trade-off between prediction accuracy and calibration. This was considered important and the first of its kind. The paper was intensively discussed among the reviewers. In particular, the reviewers argued whether the paper has shown convincingly enough that the method is necessary, because earlier results indicate other methods may produce calibrated results without the added regularizer (Haider et al. 2018). However, the results the authors point at in their response indicate a positive result, which the authors clarified in their anonymous email. The authors are strongly requested to include the additional results in their paper, as this was the bottleneck issue in recommending acceptance, and to take into account the other important points the reviewers raised. Given the other raised concerns, the paper is still very close to borderline. Finally, I would like to give special thanks to the reviewers who have made an outstanding job in evaluating this paper and taking the feedback from the authors into account. This raises our confidence in the review process which is burdened by the huge volumes at the moment.